# Link between Financial Management Behaviours and Quality of Relationship and Overall Life Satisfaction among Married and Cohabiting Couples: Insights from Application of Artificial Neural Networks

**DOI:** 10.3390/ijerph17041190

**Published:** 2020-02-13

**Authors:** Monika Baryła-Matejczuk, Viktorija Skvarciany, Andrzej Cwynar, Wiesław Poleszak, Wiktor Cwynar

**Affiliations:** 1Institute of Psychology and Human Sciences, University of Economics and Innovation, 20-209 Lublin, Poland; wieslaw.poleszak@wsei.lublin.pl; 2Faculty of Business Management, Vilnius Gediminas Technical University, LT-10223 Vilnius, Lithuania; viktorija.skvarciany@vgtu.lt; 3Institute of Public Administration, Business and Management, University of Economics and Innovation, 20-209 Lublin, Poland; andrzej.cwynar@wsei.lublin.pl (A.C.); wiktor.cwynar@wsei.lublin.pl (W.C.)

**Keywords:** financial management behaviours, overall life satisfaction, subjective well-being, quality of relationships, relationship satisfaction, shared goals and values, harsh start-up, artificial neural network (ANN)

## Abstract

Background: To explain the link between household finances and the quality of the relationship between married or cohabitating partners and their life satisfaction, the Family Stress Model (FSM) was used and placed within the theoretical framework of the Couples and Finances Theory (CFT). Methods: The measures used to examine the relationship between partners were the Financial Management Behaviour Scale, the Marriage Questionnaire (KDM-2) adapted to a version for cohabitating couples, The Shared Goals and Values Scale, Harsh Start-up Scale, and the Satisfaction With Life Scale (SWLS). In order to find out the relationship between variables, artificial neural networks (ANN) were applied. The research was conducted on a sample of 500 couples living in Poland (384 married and 116 cohabitating couples). Results: The results indicate that overall life satisfaction is most influenced by fundamental, direct, current ways of dealing with the daily financial routine and by saving and investing behaviours. Credit management and insurance behaviours are the most important for the quality of the relationship between partners. Conclusions: The research shows that financial management behaviours have an impact on the quality of relationships as well as on the subjective well-being of people in a relationship, and their relationship dynamics. This finding may be used to highlight the psychological importance of financial management behaviours.

## 1. Introduction

Various aspects of intra-household financial life have become the subject of many discussions, estimates, and analyses for many researchers and practitioners around the world who are concerned with both financial and relationship issues. They indicate the need to analyse the relationship between variables related to these following two domains: financial and relationship [1]. Overall, the conclusion is that it is essential to attain an improved understanding of the significance and role of financial management in the quality and durability of the relationships of married and cohabiting couples (e.g., [1,2,3,4,5,6]).

An attempt to explain this link is undertaken by applying the Couples and Finances Theory (CFT) model developed by Archuleta [7]. The primary assumption of the CFT theory is that financial difficulties are related to problems in the relationship ([8] as cited in [9]). The CFT is based on ecological theory, a systemic approach to investigating relationships with the financial process in the centre (every component that cooperates in bi-directional relationships). In theory, the pair system consists of a husband and wife (H&W), their marital quality (MQ), and relationship characteristics (CRC), and the financial process comprising financial inputs (FI) and financial management practices (FMP). In this article, the CFT approach is used to help to clarify the connections between the relationship of the couple and the household financial processes [7]. Additionally, in this study, the analysis was extended to include overall life satisfaction.

Another of the proposed approaches used to explain the link between intra-household finances and the quality of a relationship is the family stress model of economic pressure and marital distress, or simply the family stress model [10,11]. According to this model, negative economic events lead to economic pressure, resulting in changes to the affective states and, finally, to a decline in marital quality. This theory was supported by studies conducted in the USA in the 1980s, which indicated that negative economic events were associated with an increased sense of economic pressure, which in turn, was linked to affective changes (including increased depression and hostility that increased marital distress). Furthermore, some studies have produced evidence that support through affection as well as conflict management skills, helped (indirectly) to reduce the effects of economic pressure, and increased the odds of marital quality improvement [10] ([12] as cited in [1]). 

## 2. Literature Review

Studies led by Kerkmann, Lee, Lown, and Allgood [13] indicate that financial issues affect marital satisfaction. The authors showed that financial factors might explain 15% of marital satisfaction. However, they point out that these results should be treated with caution, as the sample consisted of young couples with a short marital relationship [13].

Other studies have drawn attention to the role played by financial arrangements, especially arguments concerning money, on marital quality. Research has shown that financial issues are an essential cause of conflict between spouses. The results obtained by Britt and Huston [14] suggest that arguments concerning money are an essential indicator of relationship satisfaction, but they do not have such a significant impact on the likelihood of divorce. However, poor financial management may harm the quality of a relationship. The consequences of bad financial management such as excessive consumer debt are related to both marital conflicts and the likelihood of divorce [15]. The results of a study by Dew and Xiao [11] suggest that financial declines are not directly related to proper financial management.

On the other hand, correct financial management is positively associated with happiness in marriages and cohabitation relationships. Furthermore, proper financial management has a direct influence on the relationship between economic pressure and relationship happiness. It also influences the relationship between financial decline and the happiness of a couple [11].

Britt, Grable, Nelson-Goff, and White [3] evaluated how the perceived own money-spending behaviours, the conduct of a partner, and the joint financial behaviour of the couple affect the degree of satisfaction experienced within relationships. The results indicated that the behaviour of the partner related to expenditures affected the degree of satisfaction derived from the relationship and the decision to stay within the relationship. Interestingly, the perceived own behaviours or common spending behaviours were not a significant factor in the quality of the relationship. Self-esteem and financial stressors were also important factors for the degree of satisfaction experienced within the relationship [3]. Archuleta, Britt, Tonn, and Grable [2] investigated the link between financial satisfaction and financial stressors and the decision of the spouse to remain married or to leave their partner. The role of demographic variables, socio-economic variables, religiosity, psychological constructs, financial satisfaction, and financial stressors as factors relevant to marital satisfaction were analysed. Religiousness and financial satisfaction were positively correlated with marital satisfaction. There was also a negative relationship between financial satisfaction and financial stressors. When the spouses experienced more financial stressors, they were also more likely to leave the marriage. The authors concluded that financially satisfied spouses were also more content with their marriages, or less willing to leave them [2].

The studies discussed above indicate the role of bad financial management as an essential stressor in relationships. Behaviours that help individuals and families to attain a more stable financial position are associated to a significant extent with a sense of satisfaction derived from the relationship [1,2,11,16]. Financial problems affect the ability of a couple to communicate and resolve conflicts and thus remain in a relationship. Both communication and financial resources are essential factors in understanding the causes of arguments between partners [17]. Compared to other types of marital misunderstandings, conflicts related to finances are more problematic for couples and are one of the best predictors of negative conflict tactics [18]. Contemporary research [19] provides an empirical basis for the development of a theoretical framework for understanding patterns of marital interactions and the impact of these patterns on marital satisfaction. The way in which couples communicate concerning financial matters is essential. Research shows that even though it is commonly believed that money is not the most frequently discussed problem within marriages, arguments concerning money are generally the most intense disagreements within married couples [19]. When couples engage in negative interactions, conflict resolution becomes more and more difficult, and marital satisfaction decreases.

However, when the partners share a sense of meaning within the relationship, their marital satisfaction increases. Couples who engage in discussions using criticism or sarcasm (i.e., forms of contempt for partners) tend to face disagreement/arguments more often. In other words, when one of the partners enters a discussion by blaming the other or criticizing, they get involved in sharp or harsh start-ups [20]. Archuleta [7] and Archuleta et al. [1] adopted the concept of shared goals and values from the work of Gottman [20]. Archuleta and colleagues reported that those who were more satisfied financially engaged less in harsh start-ups and had more shared goals and values. Additionally, positive discussion and shared goals and values were positively associated with relationship satisfaction [1]. Rosenblatt and Keller [21] found that couples who experienced more significant economic distress reported a greater degree of blaming behaviour within the marriage. The authors conclude that the economic problems of farm couples with greater economic vulnerability produce stress in these relationships.

There is evidence to suggest that healthy financial management is associated not only with marital satisfaction, but also with general life satisfaction (cf. [22]). It has been well documented that expressing healthy financial behaviours is positively related to overall life satisfaction [23] and emotional well-being [24]. Overall life satisfaction is often linked to the availability of financial resources [25,26]. On one hand, lower wages, inadequate financial management, inferior financial situation, and the general conditions associated with poverty mean that people do not have sufficient funds to pay for investments that would bring them a greater sense of satisfaction. This leads to a lower level of general life satisfaction. People who live in poverty are the most vulnerable to a sense of dissatisfaction with life. Research concerning the determinants of life satisfaction in a poor community [26] shows that for the achievement of satisfaction from life, the relevant factors include income level, employment status, or poverty status. Other studies, however, have shown that a growing income, resulting in higher purchasing power, optimism, and satisfaction, may not lead to positive changes in general life satisfaction (subjective well-being) [22].

Additionally, it has been found that excessive debt, which may arise as a result of unhealthy behaviours, is negatively linked to overall life satisfaction [27,28] and positively related to anxiety [1]. Tay et al. [29] indicated two channels through which debt may affect overall well-being. In the bottom-up spillover view, financial management (including credit management) may have considerable spillover effects in other life domains (e.g., the marriage-related indicators). On the other hand, from the resource perspective, debt imposes constraints on financial resources and, therefore, reduces the available stress buffer. According to the above-cited research, it seems that money is not the most discussed problem within marriages. However, given the intensity of arguments about money within marriages, we assumed that financial management behaviours are related to harsh start-ups, and to beliefs about shared goals and values (the meaning of money and how it should be used, the function of autonomy and independence, and with the hopes and aspirations for the family and future relationship goals).

The research conducted so far also indicates the role of unhealthy financial management behaviours as a significant stressor in the relationship, which is associated with perceived satisfaction within relationships. Additionally, spouses experiencing more financial stressors were also more likely to leave the marriage, and the consequences of unhealthy management are associated with both marital conflict and the likelihood of divorce. Therefore, it may be concluded that behaviours related to financial management are also directly related to the quality of the relationships built. 

Finally, it has been well documented that expressing healthy financial behaviours is positively related to overall life satisfaction. The debt-related financial management dimension is of particular importance. It has been established that this is an aspect of financial life that is crucial for stress and its consequences, which may be experienced in the form of mental health problems, a lower quality of social functioning, and lower global cognitive judgments of life satisfaction.

## 3. Materials and Methods

### 3.1. Financial Management Behaviour

In order to examine financial management behaviour, we adopted the financial management behaviour scale (FBMS) introduced by [30]. This is the only multi-dimensional, psychometrically validated scale that has been validated in a nationally representative sample designed to date [30]. Multi-dimensionality means that the scale—as opposed to other scales present in the relevant literature—captures (as subscales) all possible domains of household financial matters: cash management, credit management, savings and investment, and insurance. The scale of [30] is based on a consensus regarding what should be deemed as healthy financial management behaviour, which is present in the household finance literature. For instance, timely repayment of credit card debt is considered to be healthy, while not saving for retirement is unhealthy. In fact, healthy financial management behaviour follows the rules of common sense. In order to obtain data concerning financial management behaviour, the respondents were required to answer the following question: ‘On a scale from 1 (never) to 5 (always) indicate how often you have engaged in the following activities in the past six months’ (see [30] for the exact wording of the items comprising the FMBS). As a result, each subscale and the aggregate FMBS can easily be interpreted: the higher the value on the scale, the more healthy the financial behaviour.

### 3.2. Shared Goals and Values

The Shared Goals and Values Scale [7] is a four-item measurement adapted by Archuleta derived from Gottman’s [31] Shared Meaning Roles, Shared Meaning Goals, and Shared Meaning Symbols scales that are used to assess the shared meaning of couples concerning financial goals and values, life goals, and autonomy. The responses were measured using a 7-point Likert-type scale, where 1 = strongly disagree, and 7 = strongly agree. Response scores could range from 4 to 28, with lower scores indicating a lower degree of agreement concerning life goals and values, and higher scores reflecting more agreement on these issues. 

### 3.3. Harsh Start-Up

Harsh start-ups were measured using a scale consisting of five items. The scale was adapted from work originally published by Gottman [31] and translated into Polish. Conceptually, a harsh start-up may be viewed as the way in which couples interact; more specifically, how couples engage in the discussion process covering conflictual topics. Each of the following items was assessed dichotomously, with a true statement being assigned a score of 1 or 0. Items were reverse coded and summed into a harsh start-up index scale score so that higher scores reflected being less likely to engage in the harsh start-up.

### 3.4. Relationship Quality

The Well-Matched Marriage/KDM-2 questionnaire in [32] was used to measure the quality of the relationship from the perspective of four dimensions: intimacy, disappointment, self-realisation, and similarity as well as the overall result indicating overall satisfaction with the marriage/relationship. The tool has satisfactory psychometric indicators concerning research on the population of Polish marriages and couples. Cronbach’s alpha for individual sub-scales ranges from 0.81–0.89. It is the only psychometrically validated scale that has been validated in a nationally representative sample scale that has been designed in Poland to date. The KDM-2 questionnaire applies to both spouses individually and to marriages. In this research, the questionnaire was adopted to also examine cohabitation relationships and consists of 32 statements. The respondent, while answering the questions, is asked to choose one of five answers on a scale from 1 = totally disagree to 5 = totally agree.

### 3.5. Overall Life Satisfaction

In order to assess overall life satisfaction, the SWLS scale was used [33]. The scale contains five statements. Respondents assess to what extent each of them relates to their lives. The result of the measurement is a general indicator of a sense of life satisfaction, specifically global cognitive judgments of satisfaction with one’s life [34]. The SWLS asked the respondents to rate on a 5-point Likert-type scale (where 1 states for “strongly disagree” and 5 for “strongly agree”), the extent to which they agree with the five statements, for example, “In most ways, my life is close to my ideal”; “The conditions of my life are excellent”; and “If I would live my life over, I would change almost nothing”. Some researchers have shown [35] that self-satisfaction is an important component of life satisfaction and equates to well-being with high self-esteem. The Polish translation of SWLS has been used and has shown to have strong internal reliability.

### 3.6. Study Design and Sampling

The research was conducted on a sample of 500 couples living in Poland: 768 spouses and 232 cohabitants. The sample selection procedure was commissioned via a professional market and opinion research agency, DRB Research in Poland. The sample was selected using a stratified sampling technique. Specifically, the respondents were selected from different voivodeships in proportion to the number of cohabitants and marriages living therein, with control for age and education. In order to find out the relationship between financial management behaviour indicators and indicators of both marital quality and satisfaction with overall well-being, artificial neural networks (ANN) were applied. Scholars from different science fields adapted ANNs as the tool for various areas of science including social sciences. Artificial neural networks are a powerful modelling technique for indicating the relationships between variables [36].

The method of using artificial neural networks (ANNs) is based on simulating the function of the nervous system of the human brain [37,38]. The neurons summarise the impulses sent by independent variables by weight and then transfers the integral impulse to define a dependent variable [39]. In other words, ANN is a technique based on artificial intelligence and has advantages against more traditional approaches such as regression [40,41]. The first benefit is that ANNs can detect both linear and nonlinear relationships; the second, the estimation accuracy of ANNs do not depend on assumptions about the distribution of the variables [40]. These ANN characteristics are crucial for selecting this approach in determining relationships. The variables used in the study are presented in Table 1. 

The respondents had to evaluate a different number of statements in order to assess the study variables. To evaluate the statements (except harsh start-up), a 7-point Likert scale was used. In fact, the Likert scale is one of the most frequently used scales for gathering data in the social sciences [42,43]. The research model is presented in Figure 1.

The analytical model (Figure 1) seeks to investigate the relationships between financial management behaviour as represented by four variables (cash management, savings and investments, credit management, and insurance) and relationship quality, overall life satisfaction, harsh start-ups, and shared goals and values. Hence, four different models were used.

## 4. Results

In order to test the analytical model, neural networks for four distinguished models were developed (Figure 2). The number of cases used for training varied from 231 to 245, while for testing, it was from 95 to 108 (i.e., it is approximately 70/30 division in all the models, Table A1). The number of excluded cases was 661, which means that only fully-completed cases were used for the research. As a result, artificial neural networks were developed using 340 valid cases. In Figure 2, the neural networks for the FMB group variables are presented.

As can be seen in Figure 1, all models have three layers. The input layer is represented by the factors, namely cash management, savings and investments, credit management, and insurance and one output layer represents relationship quality in Model A, overall life satisfaction in Model B, harsh start-up in Model C, and shared goals and values in Model D. Moreover, all models include one hidden layer. It is worth mentioning that the grey lines show a positive relationship, while the blue shows negative relationships. Moreover, the thickness of the line shows the strength of the connection. The parameters of the weights for Model A are provided in Table 2.

All of the synaptic weights presented in Table 2 are moderate. The negative weights varied from −0.345 to −0.078, while the positive ones varied from 0.111 to 0.229, which confirms that *CashMan*, *SavInv*, *CreditMan*, and *Ins* are the variables that affect the output variable *RQ*. In order to find out which of the independent variables were the most influential when determining the value of an output variable, the importance of the variables was calculated. The results are presented in Table 3.

The importance values show that the most critical value is credit management, the second is insurance, the third is savings and investment, and the fourth is cash management. Credit management, which relates to the actions taken by households to deal with borrowed funds, is of particular importance here. Insurance-related financial behaviour is in second place in terms of affecting the quality of the relationship. 

In third place, in terms of importance, we have savings and investment. Healthy financial behaviour in this dimension is also essential for assessing satisfaction within the relationship. Finally, in terms of importance for the RQs are behaviours associated with cash management. Money management, in this case, means that the couple keeps a financial record (mental or written), maintains the discipline to stay within their budgets when spending, and also engage in comparison shopping. However, the importance of the *CashMan* variable is quite low; hence, it may be stated that it does not have a strong connection with relationship quality. In Table 4, the parameters of Model B are provided.

Table 4 shows that the negative weights varied from −0.824 to −0.250, while the positive ones varied from 0.203 to 0.495. In this case, all of the relationships can be treated as moderate or strong, but despite this fact, it showed that *CashMan*, *SavInv*, *CreditMan*, and *Ins* are the variables that have connections with the dependent variable of overall life satisfaction. In order to find out which of the independent variables were the most influential when determining the value of an output variable, the importance of the variables was calculated. The results are presented in Table 5.

Table 5 shows that the most crucial variable that relates to couples’ overall life satisfaction is cash management (the importance was 0.424 and savings and investments was 0.248). The importance of insurance and credit management were less critical than cash management. While they were less critical, the behaviours associated with insurance and savings and investments are still significant in the assessment of overall life satisfaction. This would indicate a less critical role of behaviours related to the distant future, and the greater importance of everyday financial tasks related to household tasks.

Consequently, as in the case of relationship quality, the analysis of the relationship between financial management behaviours and overall life satisfaction also indicates a significant connection. It may be concluded that healthy financial behaviour is conducive to favourable global cognitive judgments of satisfaction with the life of a spouse. It may also be concluded that another area of life, overall well-being, is affected by how couples manage their finances.

The third model to test was Model C, in which the nature of the link between cash management, savings and investments, credit management, insurance variables, and the harsh start-up variable were tested. The parameters are denoted in Table 6.

From the information presented in Table 6, the most considerable weight was assigned to the savings and investment variable, which shows that the relationship with harsh start-up is quite strong. The importance of each of the analysed variables is summarized in Table 7.

It may be observed in Table 7 that savings and investments appeared to be the most crucial variable. Credit management and insurance were also essential, while cash management was not as vital as the previously mentioned variables. The results of the calculations indicate that unhealthy behaviour in the area of savings and investment, the lack of actions taken by households to deal with borrowed funds, and the lack of behaviour aimed at protecting a contingency affects engagement more in the harsh start-up.

The fourth model, Model D, was investigated in order to determine the relationship between FMB variables (*CashMan*, *SavInv*, *CreditMan*, *Ins*) and common goals and values (*ShareGVal*). The weights of the parameters are provided in Table 8.

It may be observed in Table 8 that almost all of the weights were quite strong, which means that the relationships were also strong. However, the strongest one was the credit management weight. In order to find out which of the independent variables were the most influential when determining the value of an output variable, the importance of the variables was calculated. The results are presented in Table 9.

Table 9 shows that the most crucial variable that relates to the shared goals and values of couples were actions taken by households to deal with borrowed funds. Insurance is in second place, while cash management and savings and investments may be treated as non-essential variables, as the importance score was very low. For the shared meaning of couples concerning financial goals and values, life goals, and autonomy, the most crucial variable was credit management. Healthy behaviour associated with this financial activity is related to everyday household tasks and it turns out to be relatively unimportant for the interactional dynamics of couples. In second place in terms of importance were the behaviours related to insurance. People engaged in maintaining or purchasing an adequate health insurance policy, property insurance, and life insurance are less likely to start a conflict (they engage less in harsh start-ups).

## 5. Discussion

In this article, we attempted to address the question concerning the association between and among the various financial management behaviours, the interactional dynamics of couples, their relationship satisfaction, and overall life satisfaction. The investigation of the relationship between these variables is vital from the practical perspective of developing the acquired knowledge as well as filling the gap in the literature concerning the dimensions of healthy financial management behaviour as a factor that can protect the quality of relationships and overall quality of life. Due to the growing interest in research into the area of links between financial behaviour, the stress of couples, the dynamics of their relationships, and marital/relationship satisfaction, an attempt was made to capture the relationship between three critical areas of human functioning: psychological, financial, and social. The purpose of the analysis was to broaden the knowledge of employees of marriage counselling centres, marriage and family therapists, financial management specialists, and point out that finances do not just concern money, thus highlighting the crucial role of education and teaching competencies essential for healthy financial management. Finally, it emphasises the importance of healthy financial management dimensions for the healthy functioning of relationships and overall quality of life.

In the research conducted, the direct relationship between FMB and the quality of the relationship was first analysed. The investigation carried out indicates that the ways in which couples manage their finances affect the quality of their relationships. In other words, healthy behaviours related to achieving financial and economic goals are essential to the quality of the relationships built. The results are consistent with the research conducted to date, which indicates that the quality of relationships is related to the efficiency of their financial management (cf. [1,2,11,16]). They also indirectly point to the approach adopted throughout the research that the financial area is the foundation of the ability to meet many needs. The results of the research conducted support the concept of the Couples and Finances Theory, which in a simplified form, states that financial difficulties are related to relationship-related problems [7,8]. Similar conclusions were reached by Kerkmann, Lee, Lown, and Allgood [13], indicating that financial issues affect marital satisfaction as well as Dew and Xiao [11], who demonstrated that healthy financial management is positively associated with happiness in marriages and cohabitation. Therefore, it may be concluded that financially fit people are also more stable in their marriages (cf. [2]).

Our research shows that credit management and insurance are of the most significant importance for the quality of relationships. The results obtained broaden the knowledge about aspects of financial life related to relationship satisfaction. Healthy credit management behaviour has a positive impact on the cognitive assessment of relationship quality. Therefore, it may be assumed that healthy behaviour in this area is perceived by partners as desirable and comforting (cf. [15]). Insurance, in turn, involves far-reaching goals and protecting an unfavourable contingency. Again, it may be referred to as providing a sense of security, which is one of the basic needs of an individual. People who care about having optional insurance (against serious illness, insurance of their assets, a life insurance policy) assess their relationships better. Psychologically, this can mean that we meet basic needs (like the sense of security) and other needs are based on our financial means. That is, the way of managing and satisfying one’s needs (individual and relationship needs) no longer literally depends on obtaining various goods and services, but also depends on acquiring them through the appropriate management of financial resources.

A significant result obtained in this study is the positive relationship between cash management and engaging in harsh start-up (which is one of the dimensions of relationship dynamics). The result indicates that people with healthy behaviour in this area (who compare products, pay bills on time, stick to their budget) are less likely to get involved in the harsh start-up. Since cash management concerns basic needs, satisfying them (as important ones) will always be associated with strong emotions. Intense emotions, combined with negative interaction dynamics or a lack of conflict resolution skills, can lead to a release of tension. Displaying confidence in one’s financial management competence can be a source of compensation for relationship difficulties. These are areas worth covering in subsequent studies. The relationship between financial management and individual communication skills as well as conflict resolution in pairs would require unique clarification, and it would also be interesting to include personality factors in future research.

The way in which financial management behaviour affects the interactional dynamics of couples was also analysed. It was assumed that the aspects of relationship in a heterosexual couple that are related to communication strategies and shared goals and values in these relationships would both be directly related to relationship quality. The research conducted so far indicates that financial resources and the way they are managed are essential components for understanding the causes of arguments in a relationship [17], which in turn are significant for the quality of relationships. Research shows that if a discussion in a relationship begins with a difficult start, it will inevitably end negatively. Statistics indicate that in 96 per cent of cases, the outcome of a conversation can be predicted based on the first three minutes of a 15-min interaction. A problematic start (harsh start-up) is associated with more misunderstandings and less satisfaction within the relationship [44]. Compared to other types of marital misunderstandings, financial disagreements are more problematic for couples and are one of the best predictors of conflict tactics [18]. Quarrels about money, compared to other subjects, are the most intense disagreements in marriages (cf. [19]). In turn, similar views on (a) the importance of money and how it should be used; (b) the functions of autonomy and independence; and (c) hopes and aspirations for the family, and the future goals of relationships are related to both the relationship quality and the manner of interaction within the relationship. They are also influenced by the nature of financial management [7,8,31]. It has been documented that the way in which couples start a discussion (e.g., blaming or criticising a spouse or partner, and engaging in sharp start-ups,)also indicates that the couple has relatively few common goals and values. 

Money management and insurance are of the most significant importance for setting shared goals and values, but savings and investment and credit management are also crucial for the healthy functioning of the relationship. Healthy behaviour in the area of credit management, saving for long-term goals, putting money aside as a deferred payday, or putting it into a retirement account has a positive impact on avoiding quarrels. In turn, healthy cash management as well as insurance for the future has a positive impact on common goals and values. In other words, healthy financial management behaviours strengthen the common approach to financial matters and decrease the frequency of engagement in a harsh start-up.

The relationship between financial management behaviours and overall life satisfaction was subsequently analysed. In the context of well-being, the most influential are cash management and saving and investment behaviour. Once again, cash management plays a vital role in the quality of functioning, and efficient functioning in this dimension can be treated as a factor that protects life satisfaction. According to researchers in this area, there is a fundamental difference between concepts that measure experienced well-being and concepts that measure life evaluation [22,45,46]. Therefore, the question is, how important is financial management in how partners are satisfied with their lives as a whole. First, in light of our results, for that aspect of well-being, the most critical factor is dealing with cash. As with other aspects of relationship interactions, these fundamental behaviours are the most influential. All in all, the results of our research confirm that expressing healthy financial behaviour is positively related to overall life satisfaction [23] and emotional well-being [24]. 

## 6. Conclusions, Limitations, and Future Research

This research shows that financial management behaviour has an impact on the quality of relationships as well as on the subjective well-being of the people in the relationship.

Money management and savings and investment behaviour are the most important for the subjective quality of life. Overall life satisfaction is influenced by the most fundamental, direct, and current ways of dealing with the daily financial routine and also by regular saving and generally planning for the future (setting long-term goals including retirement). 

Overall life satisfaction is dependent on those dimensions of financial management that are associated with the daily aspects of life. These dimensions have a close psychological connection with the daily satisfaction of basic needs.

Credit management is the most critical aspect of financial management for the quality of the relationship. This finding may highlight the psychological importance of the sense of security. Healthy credit management behaviour (e.g., regular debt repayment, or comparing loan offerings) has a positive impact on how couples view their relationship. In other words, far-reaching goals are discussed (or argued about) more often than everyday shopping tasks. Insurance behaviour is also essential for the satisfaction of the relationship. Behaviours associated with non-compulsory insurance are designed to protect against unexpected changes, so they contribute toward ensuring safety. A feature of this dimension of financial management is that the insurance we studied is not mandatory, so it involves considering the future and the decision must be taken to take care of the future, both one’s own and the family’s (e.g., life insurance). Therefore, it can be said that what protects the quality of marriage from a financial perspective is the consideration of the future.

While living in cohabitation or marriage, individual needs are considered along with those in the relational context. Spouses discuss them, making decisions or at least exchanging information about them. Theoretically speaking, the quality of relationships should be associated with having common goals and values. Research shows that when people share goals and values, it is easier to maintain a consensus in this dimension and hence enjoy higher-quality relationships. This would explain why caring about far-reaching goals (such as insurance) is essential. Setting goals for the future is related to the need for security. It assumes a certain surplus and the security resulting from it. Having these skills (cash management, taking care of the future) protects against descending into conflict, which is not directly related to proper money management.

In conclusion, a list of protective factors and risks may be selected. The skilful use of money, spending it sensibly in everyday routine area translates into satisfying basic needs, and as a consequence, translates into life satisfaction and thinking about shared goals and values, which are essential for the quality of relationships. Credit management skills have a similar significance for the assessment of relationship satisfaction, overshadowing the elementary need for security. Skills in financial management are essential for the dynamics of relationships; the ability to secure the future has a unique role.

As usual, this research also has some limitations. In studies conducted to date, financial satisfaction was included in the analysis of the relationship between financial matters and marital quality as another potentially important factor. In future studies, it would be worth describing the relationship using this variable. Another topic worth exploring is the comparison between partners in relationships, as perceived by own behaviour or joint financial behaviour. It may be assumed that people in a relationship may have different perceptions of their behaviour as well as financial situations. 

Additionally, it is worth noting which psychological mechanisms underlie the combination of management skills with the quality of the relationships, for which specific aspects of the FMB relationship matter. 

Further research into the system approach would require the consideration of the role of children in households. Studies show [47] that in the case of money management, younger adults without children more often had independent money management systems. This would indicate the need to take account of having children and their importance in managing money (e.g., children introduce new categories of financial obligations).

Developing the issues discussed in the article with a positive psychology approach would indicate the need to identify both personal and social resources conducive to efficient financial management. The results are part of an already developed strategy regarding the importance of private resources for developing the quality of life.

The last of the proposed directions for research development would be to check how the method of money management, or rather the dynamics of management changes are related to the dynamics of the quality of marriage and family life. Longitudinal research concerning this topic would raise the issue of unexplored content.

The results obtained may be used in education and psychoeducation. The conclusions provide direction for preventive strategies that may be taken to support the well-being of individualsand provide them with satisfaction in their social lives. The application of the results is possible from both a psychological and economic point of view. Paying attention to the role of finance in life, the need for a good knowledge of the partner, resources, features, goals, and values before starting a relationship can help to develop it in a more satisfying manner. Appropriately teaching children for their age and developmental stage concerning money management is a protective factor for their well-being in life.

In summary, one should highlight the importance of financial management in aspects related to relationships and quality of life. The level of awareness of this should be raised for specialists working to improve the quality of marital and cohabitant relations, and above all, the psychological understanding of money should be stressed as an element of satisfying needs. The results obtained should also be considered in the context of couples and families in crisis who are searching for explanations for their situation; deficiencies in management skills (or asymmetrical competences in this aspect of the relationship) as potential causes of difficulties within the relationship and life. This knowledge may be particularly useful for psychotherapists of couples and families.

## Figures and Tables

**Figure 1 ijerph-17-01190-f001:**
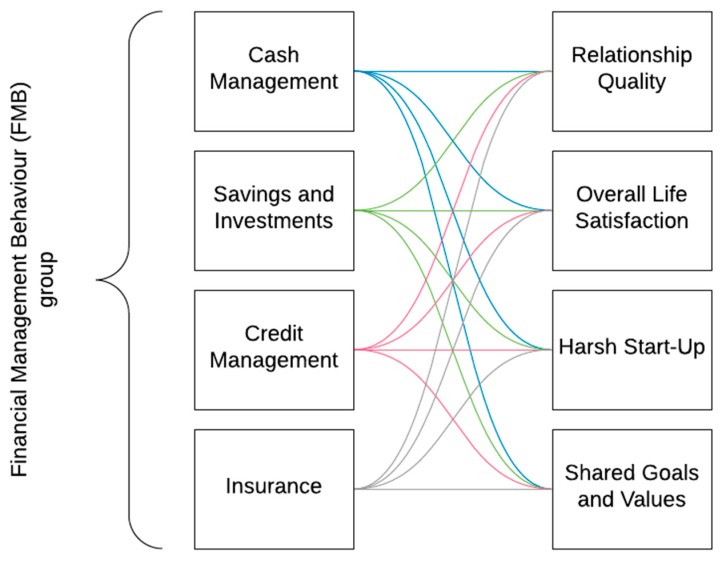
Analytical model (developed by the authors).

**Figure 2 ijerph-17-01190-f002:**
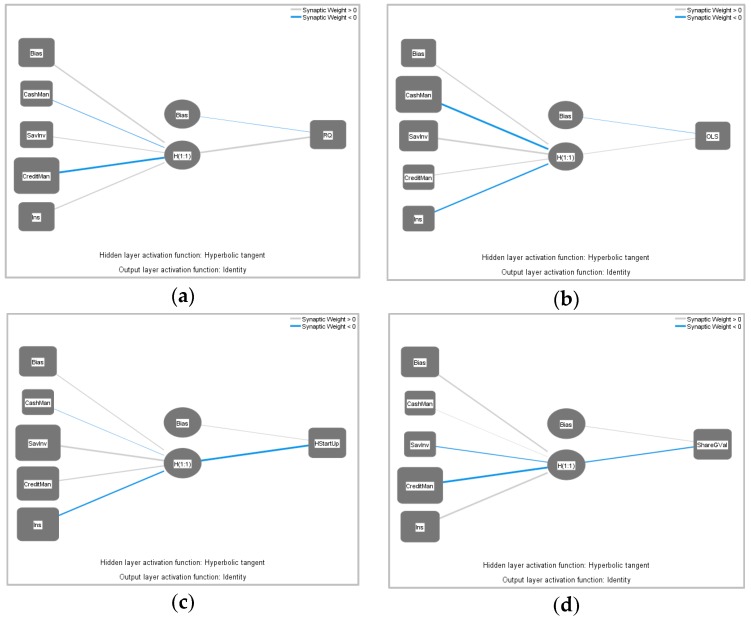
Neural network diagrams for: (**a**) Model A; (**b**) Model B; (**c**) Model C; (**d**) Model D (designed by the authors).

**Table 1 ijerph-17-01190-t001:** Variables used in the study (designed by the authors).

Variable	Number of Statements	Notation
Cash Management	Four statements	CashMan
Saving and Investments	Five statements	SavInv
Credit Management	Seven statements	*CreditMan*
Insurance	Three statements	*Ins*
Relationship Quality	Thirty-two statements	*RQ*
Overall Life Satisfaction	Five statements	*OLS*
Harsh Start-Up	Five statements	*HStartUp*
Shared Goals and Values	Four statements	*ShareGVal*

**Table 2 ijerph-17-01190-t002:** Parameter estimates for Model A (calculated by the authors).

Predicted			
		**Hidden Layer 1**	**Output Layer**
**Predictor**		H (1:1)	RQ
**Input Layer**	(Bias)	0.229	
	CashMan	−0.078	
	SavInv	0.111	
	CreditMan	−0.345	
	Ins	0.174	
**Hidden Layer 1**	(Bias)		−0.009
	H (1:1)		0.287

**Table 3 ijerph-17-01190-t003:** Independent variable importance for Model A (calculated by the authors).

FMB Group Variables	Importance	Normalised Importance
Cash Management	0.140	29.8%
Savings and Investments	0.162	34.5%
Credit Management	0.468	100.0%
Insurance	0.230	49.2%

**Table 4 ijerph-17-01190-t004:** Parameter estimates for Model B (calculated by the authors).

Predicted			
		**Hidden Layer 1**	**Output Layer**
**Predictor**		H (1:1)	OLS
**Input Layer**	(Bias)	0.249	
	CashMan	−0.824	
	SavInv	0.495	
	CreditMan	0.203	
	Ins	−0.250	
**Hidden Layer 1**	(Bias)		−0.021
	H (1:1)		0.191

**Table 5 ijerph-17-01190-t005:** Independent variable importance for Model B (calculated by the authors).

FMB Group Variables	Importance	Normalised Importance
Cash Management	0.424	100.0%
Savings and Investments	0.284	66.9%
Credit Management	0.143	33.8%
Insurance	0.148	35.0%

**Table 6 ijerph-17-01190-t006:** Parameter Estimates for Model C (calculated by the authors).

Predicted			
		**Hidden Layer 1**	**Output Layer**
**Predictor**		H (1:1)	HStartUp
**Input Layer**	(Bias)	0.092	
	CashMan	−0.023	
	SavInv	0.279	
	CreditMan	0.213	
	Ins	−0.253	
**Hidden Layer 1**	(Bias)		0.071
	H (1:1)		−0.360

**Table 7 ijerph-17-01190-t007:** Independent variable importance for Model C (calculated by the authors).

FMB Group Variables	Importance	Normalised Importance
Cash Management	0.041	11.3%
Savings and Investments	0.367	100.0%
Credit Management	0.301	81.9%
Insurance	0.290	79.0%

**Table 8 ijerph-17-01190-t008:** Parameter estimates for Model D (calculated by the authors).

Predicted			
		**Hidden Layer 1**	**Output Layer**
**Predictor**		H (1:1)	ShareGVal
**Input Layer**	(Bias)	0.620	
	CashMan	0.007	
	SavInv	−0.126	
	CreditMan	−1.820	
	Ins	0.947	
**Hidden Layer 1**	(Bias)		0.054
	H (1:1)		−0.304

**Table 9 ijerph-17-01190-t009:** Independent variable importance for Model D (calculated by the authors).

FMB Group Variables	Importance	Normalised Importance
Cash Management	0.003	0.5%
Savings and Investments	0.046	7.5%
Credit Management	0.610	100.0%
Insurance	0.341	55.9%

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
