# Peer review of "Link between Financial Management Behaviours and Quality of Relationship and Overall Life Satisfaction among Married and Cohabiting Couples: Insights from Application of Artificial Neural Networks"

_ijerph, 2020, doi:10.3390/ijerph17041190_

Round 1

Reviewer 1 Report

This study aims to examine the link between household finances and the quality of the relationship between married or cohabitating partners and their life satisfaction using artificial neural networks on a representative sample of Polish couples. The study has important findings that are useful in the field. However, in order to improve the paper, there are several areas need to be improved before I can make decision on the paper.

Firstly, the description of the study design and sampling on how the sample were derived are not clearly explained in the paper. Authors mention that the sample is a representative sample of polish couples. However, how respondents were recruited or whether they have used one of probability sampling techniques such as simple random sampling, stratified sampling, clustering, etc currently not available in the current version. Suggest adding one section on study design and sampling to describe the process on how the sample was derived in material and method section.

Secondly, the authors decided to use artificial neural networks in their analysis. Since this type of analysis is not commonly used in medical research. I think it would be good if authors can explain further how this approach was implemented in their study. The authors can describe in more detail about this approach under the statistical section which is now is not available in the paper.

Author Response

Response to Reviewer 1 Comments

Dear Reviewer,

Thank you for the opportunity to revise our paper on ‘Link between Financial Management Behaviours and Quality of Relationship and Overall Life Satisfaction among married and cohabiting couples: Insights from Application of Artificial Neural Networks’. Your suggestions have been very helpful.

I have included your comments after this letter and responded to them indicating how we addressed each concern and describing the changes we have made. The revisions have been approved by all authors. The changes are marked in the ‘track changes’ mode in the paper.

We hope the revised manuscript will better suit and we thank you for your continued interest in our research.

Kind regards,

Monika Baryła-Matejczuk

Point 1:

Firstly, the description of the study design and sampling on how the sample were derived are not clearly explained in the paper. Authors mention that the sample is a representative sample of polish couples. However, how respondents were recruited or whether they have used one of probability sampling techniques such as simple random sampling, stratified sampling, clustering, etc. currently not available in the current version. Suggest adding one section on study design and sampling to describe the process on how the sample was derived in material and method section.

Response 1: Section 3.6 entitled “Study Design and Sampling” was added (please see below). As indicated in the new section, the sample selection procedure was commissioned to a market research agency with a request to ensure representativeness of the sample. After receiving your comment, we contacted the agency and asked it to deliver details concerning the selection procedure. As a result, the procedure is briefly described in the newly added section based on the explanation delivered by DRB Research. Although the agency maintains that the applied selection procedure ensures representativeness, we eventually decided not to use the term “representative” in the article to avoid controversies around the issue.

The research was conducted on a sample of 500 couples living in Poland: 768 spouses and 232 cohabitants. The sample selection procedure was commissioned to a professional market and opinion research agency – DRB Research in Poland. The sample was selected using stratified sampling technique. Specifically, the respondents were selected from different voivodeships in proportion to the number of cohabitants and marriages living therein, with control for age and education.

Point 2:

Secondly, the authors decided to use artificial neural networks in their analysis. Since this type of analysis is not commonly used in medical research. I think it would be good if authors can explain further how this approach was implemented in their study. The authors can describe in more detail about this approach under the statistical section which is now is not available in the paper.

Response 2: The approach is described in 3.6 section.

ANNs is a technique base on artificial intelligence and has advantages against more traditional approaches such as regression (Partovi & Anandarajan, 2002). The first benefit is that ANNs could detect both linear and nonlinear relationships, the second – the estimate accuracy of ANNs do not depend on assumptions about the distribution of the variables (Partovi & Anandarajan, 2002). These ANNs’ characteristics were crucial for selecting this approach for relationships determining.

Reviewer 2 Report

The following is my review of the manuscript, "The link between Financial Management Behaviours and Quality of Relationship and Overall Life Satisfaction among married and cohabitating couples: The insights from the application of Artificial Neural Networks.” The authors investigated “the link between household finances and the quality of the relationship between married or cohabitating partners and their life satisfaction. The Family Stress Model (FSM) was used and placed within the theoretical framework of the Couples and Finances Theory (CFT).

The paper makes a valuable contribution to the literature and the experiments are of value to the readers of "IJERPH". The paper's aims and scope match those of IJERPH; therefore, the topic is appropriate for this journal. The proposal is appealing and interesting, and the method deserves consideration. The paper, however, demonstrates an inadequate understanding of the relevant literature. I would suggest adding the following paper that also utilizes relationship satisfaction quality improvement as a baseline:

Rodger, J. A. (2014). Reinforcing Inspiration for Technology Acceptance: Improving Memory and Software Training Results Through Neuro-Physiological Performance.   Computers in Human Behavior.

The previously mentioned strengths of the manuscript suggest that the paper can be considered to be of both high quality and of sufficient value to the readership of "IJERPH", once the mentioned literature is investigated and incorporated into the manuscript. Therefore, based on the foregoing discussion, I would recommend that this paper be accepted after reviewing and integrating the above pertinent citation and revising the manuscript for improved flow and clarity.

Author Response

Response to Reviewer 2 Comments

Dear Reviewer,

Thank you for the opportunity to revise our paper on ‘Link between Financial Management Behaviours and Quality of Relationship and Overall Life Satisfaction among married and cohabiting couples: Insights from Application of Artificial Neural Networks’. Your suggestions have been very helpful for improving the manuscript.

I have included Your comment after this letter and responded to indicating how we addressed Your suggestion and describing the changes we have made. The revisions have been approved by all authors. The changes are marked in the ‘track changes’ mode in the paper.

We hope the revised manuscript will better suit and we thank you for your continued interest in our research.

Kind regards,

Monika Baryła-Matejczuk

Point 1:

The paper, however, demonstrates an inadequate understanding of the relevant literature. I would suggest adding the following paper that also utilizes relationship satisfaction quality improvement as a baseline:

Rodger, J. A. (2014). Reinforcing Inspiration for Technology Acceptance: Improving Memory and Software Training Results Through Neuro-Physiological Performance. Computers in Human Behavior.

Response 1: The proposal to literature was considered as an important alternative to the methods of analysis (ANN) used in our research. It was introduced in the research methodology section. Thank you for the suggestion of improvement and the opportunity to refer readers to other studies.